# New Insights into the Reparative Angiogenesis after Myocardial Infarction

**DOI:** 10.3390/ijms241512298

**Published:** 2023-08-01

**Authors:** Marta Martín-Bórnez, Débora Falcón, Rosario Morrugares, Geraldine Siegfried, Abdel-Majid Khatib, Juan A. Rosado, Isabel Galeano-Otero, Tarik Smani

**Affiliations:** 1Group of Cardiovascular Pathophysiology, Institute of Biomedicine of Seville, University Hospital of Virgen del Rocío/University of Seville/CSIC, Avenida Manuel Siurot s/n, 41013 Seville, Spain; mmartin55@us.es (M.M.-B.); dfalcon-ibis@us.es (D.F.); rmorrugares-ibis@us.es (R.M.); 2Department of Medical Physiology and Biophysics, Faculty of Medicine, University of Seville, 41009 Seville, Spain; 3Department of Cell Biology, Physiology and Immunology, Universidad de Córdoba, 14071 Córdoba, Spain; 4RyTME, Bordeaux Institute of Oncology (BRIC)-UMR1312 Inserm, B2 Ouest, Allée Geoffroy St Hilaire CS50023, 33615 Pessac, Francemajid.khatib@inserm.fr (A.-M.K.); 5Cellular Physiology Research Group, Department of Physiology, Institute of Molecular Pathology Biomarkers (IMPB), University of Extremadura, 10003 Caceres, Spain; jarosado@unex.es

**Keywords:** angiogenesis, endothelial cell, cardiac repair, heart infarction

## Abstract

Myocardial infarction (MI) causes massive loss of cardiac myocytes and injury to the coronary microcirculation, overwhelming the limited capacity of cardiac regeneration. Cardiac repair after MI is finely organized by complex series of procedures involving a robust angiogenic response that begins in the peri-infarcted border area of the infarcted heart, concluding with fibroblast proliferation and scar formation. Efficient neovascularization after MI limits hypertrophied myocytes and scar extent by the reduction in collagen deposition and sustains the improvement in cardiac function. Compelling evidence from animal models and classical in vitro angiogenic approaches demonstrate that a plethora of well-orchestrated signaling pathways involving Notch, Wnt, PI3K, and the modulation of intracellular Ca^2+^ concentration through ion channels, regulate angiogenesis from existing endothelial cells (ECs) and endothelial progenitor cells (EPCs) in the infarcted heart. Moreover, cardiac repair after MI involves cell-to-cell communication by paracrine/autocrine signals, mainly through the delivery of extracellular vesicles hosting pro-angiogenic proteins and non-coding RNAs, as microRNAs (miRNAs). This review highlights some general insights into signaling pathways activated under MI, focusing on the role of Ca^2+^ influx, Notch activated pathway, and miRNAs in EC activation and angiogenesis after MI.

## 1. Introduction

Coronary artery disease remains one of the main causes of mortality worldwide, being myocardial infarction (MI) the most responsible for these deaths [1,2]. In recent years, remarkable advances in treatments for MI have been achieved thanks to successful and prompt strategies of revascularization, which have led to an improvement in patients’ outcomes and survival [3]. However, adverse consequences resulting from oxygen deficiency due to the impact of ischemia, the loss of cardiomyocytes, and the resulting hypertrophy and fibrosis inevitably provoke the progress of the MI disease toward heart failure [4]. Hence, it remains urgent to search for new therapies capable to efficiently re-establish perfusion and provide oxygen and nutrients to mitigate ischemic damages. During MI, ischemia and healing cardiac tissue trigger a cascade of signaling reactions involving an increase in reactive oxygen species (ROS) and intracellular Ca^2+^ levels, a dysregulation of endothelial NO synthase (eNOS) [5,6,7], and the activation of hypoxia-induced factor 1α (HIF-1α) required for blood vessels formation from pre-existing vasculature, an action carried out by endothelial cells (ECs) known as angiogenesis [8], as summarized in Figure 1. In addition to the role of ECs in post-infraction angiogenesis, many reports demonstrated that most cell populations of the heart, including cardiomyocytes, macrophages, fibroblasts, and monocytes, participate in this process because they express a high amount of vascular endothelial growth factor (VEGF), the proangiogenic factor par excellence, up to one week after the ischemic event, and other growth factors. Complementarily, ECs close to the infarcted region show an increase in VEGF receptor 2 (VEGFR2) expression [9,10,11], which would explain ECs activation and the starting of new blood vessel formation. Other co-factors might be taken into consideration in cardiac angiogenesis given the high amount of research demonstrating an increase in pro-angiogenic paracrine/autocrine signals after MI due to the action of fibroblast growth factor 2 (FGF2), hepatocyte growth factor (HGF), platelet-derived growth factor (PDGF), insulin growth factor-1 (IGF-1) [12,13,14,15,16], hormones (estradiol, estrogen, etc.) [17,18], interleukins 2, 6, 17 (IL2, IL6, IL17), transforming necrosis factor α (TNF-α), and monocyte chemoattractant protein-1 (MCP-1) [19,20,21,22]. Likewise, other studies showed an exacerbated traffic of vesicles released by different heart cell populations whose contents include proteins and non-coding RNA, such as microRNAs (miRNAs) [23,24]. All these molecular players trigger a plethora of signaling pathways and changes in the free intracellular Ca^2+^ concentrations ([Ca^2+^]_i_), among other signaling, which fine-tune and orchestrate the complex process of angiogenesis, as illustrated in Figure 1 and recently reviewed [25]. However, the large number of signaling pathways participating in angiogenesis after MI makes it difficult to determine a clear mechanism for an optimal angiogenic response, as discussed recently [10,26]. This review provides an overview of the recent literature showing the role of different signaling pathways and miRNAs in ECs activation and angiogenesis after MI.

## 2. Reparative Role of Mature and Endothelial Progenitor Cells in the Infarcted Heart

Two different phenotypes of ECs can be distinguished in this process [27]. Tip cells are highly migrative and poorly proliferative vessel wall stalk cells, which have high proliferative and poor migrative rates [28]. Beyond these two phenotypes, a recent study using a single-cell RNA sequencing approach in mice infarcted hearts suggested the presence of up to ten clusters of heart ECs population with different gene expression signatures, suggesting the functional richness of these cells [29]. New capillary formation under MI starts from the border zone and continues to the infarcted necrotic zone, which is considered a reparative process because a correctly harmonized post-ischemic angiogenesis has been associated with a better prognosis of the disease [27,29]. Indeed, early neovascularization of an infarcted heart is supposed to reduce the infarct size and limit the risk of adverse cardiac remodeling, as well as improve cardiomyocytes’ survival and heart function, as demonstrated in the microstructure of rabbit cardiac tissue [30]. However, mature resident ECs are terminally differentiated cells with limited proliferative and migrative potentials; therefore, their role in myocardial infarction angiogenesis is still under debate [31]. However, ECs can be stimulated by a plethora of endothelial progenitor cells (EPCs)-released pro-angiogenic factors such as VEGF, FGF, or IGF, to promote new capillary formation [32]. In the meantime, EPCs are considered the precursors of blood vessels due to their robust angiogenic potential since they have high proliferative and migrative skills required for new blood vessel formation (for review, [33]). The definition of endothelial progenitors has varied widely in the literature [32], but at least two clear subgroups of EPCs have been implicated in post-ischemic neovascularization, myeloid angiogenic cells (MACs), previously known as early outgrowth cells, and endothelial colony-forming cells (ECFCs), also referred to as late EPCs or late outgrowth EPCs [33]. MACs are not able to differentiate into adult ECs, and consequently, they are not capable of forming new vessels themselves. MACs theoretically act by secreting pro-angiogenic mediators to stimulate ECs and recruit more circulating EPCs (Figure 2). For instance, stromal cell-derived factor 1 (SDF-1), through the activation of G protein-coupled receptor CXCR4, mediated bone marrow progenitor cells recruitment to ischemic cardiac tissue during permanent ligation of coronary artery in mice [34]. SDF-1 is a well-studied chemotactic cytokine whose levels increase in the infarcted zone [35], enhancing the homing of mobilized progenitor cells to areas of SDF-1 production. By contrast, ECFCs can differentiate into new mature ECs, promoting angiogenesis [36,37], which could be the most suitable cellular substrate to induce reparative angiogenesis in MI. An early study showed that the administration of ECFCs in a porcine model of AMI decreased the infarct size and increased microvessel density, which attenuated the adverse myocardial remodeling [38]. Another recent study demonstrated that the transplantation of ECFCs combined with mesenchymal stem cells (MSCs) into MI mice with permanent ligation of the left descendent coronary artery significantly improved cardiac function and alleviated the ischemic cardiac damage by increasing capillary density 30 days after AMI, although the underlying signaling pathways of this combined cell-therapy has not been addressed [39]. Likewise, a previous report indicated that the injection of ECFCs and MSCs into the infarcted heart of MI mice promoted a regenerative effect seemingly mediated by the upregulation of multiple paracrine factors associated with angiogenesis such as Ang-2, FGF-2, IGF-1, and SDF-1α [40].

Interestingly, single-cell transcriptome analysis confirmed that resident cardiac but not myeloid EPCs are the main ones responsible for ECs clonal expansion after the ischemic event [41]. In the same vein, the use of hematopoietic and vessel wall EPCs from male patients whose bone marrow transplant was owned by a female donor demonstrated that only endothelial lineages EPCs produced mature ECs [42]. Thus, although circulating EPCs seems to be a promising therapy to directly improve neovascularization after MI [43], nowadays, they are being used mainly as a vector to carry out microRNAs (miRNAs) and other mediators involved in angiogenesis [44,45]. Recently, a nice study revealed that an Ionic solution of calcium silicate bioceramics regulated extracellular vesicles (eVs) synthesis in EPCs, which, when delivered in gelatin methacryloyl-polyethylene glycol microsphere, promoted angiogenesis and improved cardiac function after MI [46]. This study further demonstrated that stimulated EPCs eVs promoted angiogenesis by transferring highly expressed miR-126a-3p to cardiac ECs, which stimulated SDF-1α and CXCR4 expression. Therefore, this study supports the idea that EPCs might require extra stimulation to secrete high-yield bioactive eVs that need to be delivered to the infarct site using stable hydrogel microspheres to effectively promote angiogenesis and heart recovery after MI. 

## 3. Signaling Pathways Participating in the Post-Ischemic Angiogenesis

The role of Notch in angiogenesis cardiac repair is still under debate. Notch pathway plays an important role in both healthy and pathological conditions since it controls angiogenesis and endothelial sprouting, among other processes. Key proteins of this pathway, namely Notch1, Notch4, Jagged1, delta-like 1 and 4 (DLL1 and DLL4), are expressed in ECs preserving their homeostasis and normal function [47]. Many studies demonstrated the role of Notch in neovascularization, stem cell differentiation and fibrosis mitigation after MI [48,49]. Notch1 and associated proteins, Hes1 and Jagged1, are significantly in-creased up to 4 days after MI in cardiomyocytes [50]. Likewise, Notch is associated with MI recovery by stimulating RBP-J signaling pathway, thus limiting ventricular remodeling, and improving cardiac function [51], and improving cardioprotection [52], but also promoting angiogenic processes, as discussed recently [48,52,53,54,55]. For instance, the ad-ministration of an adenovirus overexpressing the Notch intracellular domain (NICD) in rat’s infarcted heart increased VEGF staining and angiogenesis, improving cardiac function [52]. Recently, neovascularization 24h after MI was induced by the administration of extracellular vesicles derived from cardiac MSCs overexpressing NICD-overexpressing in mice [49]. 

Interestingly, Notch is not a solo player in post-infarction angiogenesis, other signaling pathways synergistically work together with Notch, as VEGF-A [5] or HIF-1α, during MI [56]. Indeed, HIF-1α stimulated MSCs secretion of Jagged1-containing exosomes after MI, activating ECs and inducing angiogenesis, in vitro, as assessed by capillary tube formation, as well as in vivo using a Matrigel assay in athymic nude mice [8]. Notch pathway also crosstalk with the phosphoinositide 3-kinase (PI3K)/Akt signaling pathways after MI, since Notch was activated by PI3K/Akt signaling, meanwhile, Notch itself enhanced the expression of PI3K/Akt signaling in adult myocardium following MI, suggesting a positive survival feedback mechanism between Notch and Akt signaling [50]. Interestingly, main downstream effectors of Akt signaling, eNOS, VEGF, mTOR or FOXO [57,58,59] con-tributed to myocardial angiogenesis. For example, VEGF induced Akt and FOXO3a phosphorylation, limiting its transcriptional activity and enhancing cardiomyocyte survival and native angiogenesis in the post-ischemic environment [60,61]. 

In addition to those widely studied classical pathways others like Wnt/β-Catenin and JAK/STAT also play a role in post-infarction angiogenesis. As reviewed recently, Wnt signaling is triggered after MI injury, being related to inflammation and angiogenesis [62]. In fact, protein levels of members of Wnt family (Wnt2, Wnt4, Wnt10b, and Wnt11) are increased 5 days following MI in perivascular smooth muscle actin-positive cells and subepicardial ECs, seemingly through endothelial-to-mesenchymal transition mechanism [63]. Moreover, using Wnt signaling reporter in modified mice and rats, it was demonstrated that canonical Wnt signaling increased in the border zone after MI [63,64,65]. In addition, genetically enhanced Wnt10b expression in cardiomyocytes of transgenic mouse boosted arterial formation and attenuated fibrosis in cardiac tissue after the injury [66]. In the case of the role of JAK/STAT3 pathway in neovascularization under MI, STAT3 cardiac specific KO mice exhibited reduced myocardial capillary density and more susceptibility to myocardial ischemia/reperfusion injury, even though no variations in VEGF expression was observed [67]. By contrast, STAT3 activation by Astragaloside IV, a Chinese drug herb that alleviates ischemic heart failure enhanced CD31+ stained cells (EPCs) in ischemic hearts and stimulated HUVEC-tube formation [68]. Similarly, the administration of hyaluronic acid oligosaccharides into MI mice promoted neovessels formation and improved cardiac function as assessed 28 days post-intervention. These beneficial effects were associated with the activation of chemokines expression implicated in macrophage polarization, and the stimulation of MAPK and JAK/STAT signaling pathway for myocardial function reconstruction, as revealed by transcriptomic analyses [69]. These studies suggest that JAK/STAT signaling pathway has a role in cardiac remodeling after cardiac infarction by controlling angiogenesis among others cellular processes as reviewed recently [25].

## 4. Reactive Oxygen Species Regulation of Angiogenesis

Oxidative stress can be both a cause and consequence of physiological or pathological angiogenesis regulation [70,71]. A recent study used a novel conditional binary transgenic mouse overexpressing the mitochondrial antioxidant isoform of superoxide dismutase in an ECs specific manner to demonstrate that they have increased coronary capillary and arteriolar density in the post-MI ischemic region and improved left ventricle function in MI mice model [72]. This study showed that specific reduction in ECs mitochondrial ROS induced mitochondrial complex I biogenesis, increased proteins associated with oxidative phosphorylation pathways such as COX6A1, NDUFA9, NDUFB1, NDUFV3, NDUVB3 and NDUVB7 in coronary ECs, and induced coronary angiogenesis. By contrast, early reports suggested that a well-controlled oxidative stress is beneficial for angiogenesis during tissue repair because low levels of ROS seemingly promote the production of cytokines and growth factors, such as VEGF through HIF-1α in the injured myocardium, which handled angiogenesis by cell survival, proliferation, and ECs apoptosis regulation [70,71]. ROS production in ECs can be induced by VEGF via NADPH oxidase through Nox2 subunit regulation. In this way, Nox2 knockout (KO) mice showed significant downregulation of pro-angiogenic genes and worse prognosis after MI, as compared to control mice [73]. These data suggest a fine-tune regulation of angiogenesis under oxidative stress.

## 5. Ca^2+^ Signaling in Angiogenesis

Ca^2+^ ion is an important second messenger that regulates many primary functions of ECs, as proliferation, migration, control of permeability, the synthesis and release of vasoactive factors and angiogenesis [74,75,76]. Different studies have demonstrated the contribution of several cationic channels which permeate Ca^2+^ to angiogenesis, such as transient receptor potential (TRP) and store operated Ca^2+^ (SOC) channels, using different approaches and transgenic mice [77,78,79,80,81]. Concretely, many studies demonstrated that TRPC and protein related to SOC entry, like Orai1 [82], Orai3 [83], SARAF [78], STIM1 and 2 [82,84] are involved in ECs activation and angiogenesis, but their role in MI angiogenesis has not been revealed yet. Compelling evidence demonstrated that proangiogenic growth factors, such as VEGF, increased [Ca^2+^]_i_ through the activation of non-excitable Ca^2+^ channels, modulating the signal transduction pathway leading to angiogenesis [78,85]. In the context of MI, significant increase of Orai isoforms and TRP channels has been observed in peri-infarcted hearts, 1 week after the intervention in rats [6,86]. A recent study demonstrated the role of TRPC1 in post-ischemic angiogenesis using EC-specific TRPC1 KO mice (TRPC1EC^−/−^) [77]. TRPC1EC^−/−^ mice showed more depressed cardiac function than TRPC1^fl/fl^ mice, as evaluated by a reduced ejection fraction and fractional shortening. Interestingly, the staining of ECs with anti-CD31 confirmed a significant decrease in capillary density in the infarct area of the hearts of TRPC1EC^−/−^ mice. Moreover, the expression of HIF-1α induced by dimethyloxaloylglycine stimulated TRPC1 expression in primary mouse coronary artery ECs and improved cardiac function of TRPC1^fl/fl^, as compared to TRPC1EC^−/−^ mice after MI, which was associated with an increased capillary density, decreased infarct size and improved ejection fraction. Altogether, this study concludes that TRPC1 stimulation of angiogenesis contributes to the functional recovery of post-infarcted heart [77]. Similarly, TRPC5 channel has been proposed as fair therapeutic candidate that may recover ischemic tissue through angiogenesis [87], because TRPC5 expression together with Orai1 was increased after MI, forming SOC channels [6]. TRPC5 role in angiogenesis has been proved in hind limb ischemia model [88], and an animal model of spinal cord ischemia/reperfusion (I/R) injury [89]. Nevertheless, the impact of TRPC5 on the improvement of myocardial angiogenesis has not been demonstrated. Therefore, the specific role of Ca^2+^ entry through different isoforms of TRP and SOC channels in the context of reparative angiogenesis after MI still need to be verified.

## 6. miRNAs as Regulators of Post-Ischemic Angiogenesis

miRNAs play an important role as epigenetic regulators of endothelial function, therefore they were explored as potential therapeutic targets for promoting angiogenesis to improve cardiac repair, as reviewed recently [90]. miRNAs are small non-coding RNAs (∼22 nucleotide) that regulate gene expression by binding mainly to the 3’UTR region of target mRNA. By forming imperfect base-pairing with target mRNA, miRNAs can either cleave the mRNA directly or repress its translation, allowing them to fine-tune gene expression, playing crucial roles in cellular processes such as post-MI angiogenesis [91]. Traffic of microvesicles released from the different heart cell populations containing miRNAs has been detected, supporting cell-to-cell communication under MI [92,93]. In this section, we will summarize the current state of research on miRNA-mediated regulation of angiogenesis following MI through their regulation of key pro-angiogenic signaling pathway as summarized in Table 1. 

One of the main gene regulated by miRNA in angiogenesis is *HIF-1α* which expression is known to increase after myocardial ischemia and regulates the expression of pro-angiogenic genes. An interesting study used rat model for MI to demonstrate that rodent miR-322, homolog of human miR-424, was upregulated in the peri-infarct cardiac tissues [94]. Additional experiments in ECs under hypoxia showed that miR-424 regulated the expression of HIF-1α through its proteasomal degradation process. ECs stimulation by hypoxia increased C/EBPα in conjunction with RUNX-1 to activate the PU.1 promoter, which stimulated the increase of miR-424 level. Moreover, miR-424 targeted CUL2 which destabilized the VCBCR U3-ligase complex, causing translocation of HIF-1α to the nucleus of ECs that stimulated pro-angiogenic genes transcription as *VEGF*, glucose transporter 1 (*GLUT1*), and erythropoietin (*EPO*) [94]. Interestingly, miRNAs might also inhibit angiogenesis through HIF-1α during MI. A recent study suggested that upon the insult of ischemia, exosomes including miR-19a-3p were secreted from cardiomyocytes that are absorbed by ECs, which inhibited their proliferation via HIF-1α downregulation. At the same time, the level of miR-19a-3p were increased in serum of patients with acute MI, and in culture medium of cardiac myocytes stressed with H_2_O_2_ protocol to mimic hypoxia [95]. In contrast, downregulation of miR-19a-3p promoted ECs survival and proliferation, and improved heart function by restoring HIF-1α expression. Significant increase in the expression of ECs markers, CD31, was observed in antagomiR-19a-3p transfected mice’s hearts, as compared with non-transfected group after MI, suggesting that the administration of antagomiR-19a-3p enhanced angiogenesis and improved heart function in MI mice by modulating HIF-1α expression [95]. Therefore, these studies suggested that HIF-1α can be modulated by miRNAs in ECs either positively or negatively which will affect the onset of angiogenesis after MI. 

VEGF pathway is also regulated by miRNAs in heart, as reviewed elsewhere [96]. Independent studies focused on miR-126 as the main regulator of physiological angiogenesis since it has been demonstrated that miR-126 represses negative regulators of VEGF pathway, such as sprouty-related protein 1 (SPRED1) and PI3K regulatory subunit 2 (PIK3R2). Therefore, miR-126 action enhanced ECs response to VEGF, improving angiogenesis [97]. Its implication in angiogenesis following MI also has been demonstrated by the intramyocardial injection of MSCs overexpressing miR-126 into infracted area, which enhanced microvessel formation and increased blood flow to the infarcted and peri-infarcted zones. This study also determined that this treatment increased the level of proteins involved in VEGF pathway and Notch signaling in MSCs, specifically VEGF, basic FGF and DLL4 [98]. Furthermore, the transplantation of EPCs overexpressing miR-126-3p in an ischemic cardiomyopathy rat model created by left coronary artery ligation, stimulated angiogenesis, and improved cardiac function, which correlated with the alteration of pro-angiogenic cytokine expression, as VEGF-A, IL-10, IL-3, IGF-1, and angiogenin [99]. Interestingly, an early study has shown that miR-126 and miR-130a are downregulated in angiogenic MACs from patients with chronic heart failure (CHF) [100]. Meanwhile, the administration of MACs from patients with CHF transfected with miR-126 and miR-130a in nude mice with MI improved cardiac contractility and enhanced capillary density, indicating efficient neovascularization and cardiac repair under these conditions. In vitro, those MACs transfected with miR-126 or miR-130a mimics markedly enhanced their capacity to promote HUVECs-tube formation, an action associated with *SPRED1* downregulation [100]. miR-130a seems also involved in the regulation of VEGF and PI3K/Akt signaling pathways. In fact, the administration of lentivirus expressing miR-130a into mouse hearts seven days before they were subjected to MI attenuated cardiac dysfunction and improved remodeling, which was associated with an enhanced microvascular density. Using in vitro wound healing assay, it was confirmed that miR-130a stimulated HUVEC migration, involving PTEN suppression, PIP3/Akt pathway activation, and VEGF levels’ increase [101]. *VEGF* is also a target gene of miR-375, which levels are increased both in rodent models of MI and in patients with heart failure. Knockdown of miR-375 enhanced *VEGF* expression in the infarcted myocardium, attenuated MI-induced inflammation, cardiomyocyte apoptosis and restored left ventricle function and remodeling. Moreover, the effect of miR-375 knockdown increased PDK-1 expression and downstream survival signaling AKT in cardiac ECs, cardiac myocytes, and macrophages [102]. A recent study, provided evidence showing that significant downregulation of miR-21 in post-MI hearts of mouse model injected with cardiac stromal cells from patients with heart failure. Importantly, restoring miR-21-5p expression contributed to heart repair by enhancing angiogenesis and cardiomyocyte survival through *PTEN* inhibition, which triggered the activation of Akt kinase activity and promoted the expression of VEGF in ECs [103]. PTEN signaling pathway activation of angiogenesis was also regulated by miR-499-5p stimulated by Tanshionone IIA, the main active monomer compound of Danshen, injected in the peri-infarcted mice heart, which improved cardiac function after MI by increasing the expression of VEGF and angiotensin-1, activating angiogenesis [104]. Many other miRNAs boosted neovascularization by targeting other non-classical path-ways in ECs. For example, miR-378 enhanced angiogenic capacity of CD34+ cells by targeting suppressor of fused (*SuFu*) and *Fus-1* expression, which trended to upregulate pro-angiogenesis factors [105]. By contrast, other miRNAs have anti-angiogenic role and appear reduced after MI such as miR-185-5p, miR-143 and miR-92a, which target *CatK*, IGF signaling pathway and integrin subunit alpha5 (*ITGA5*), respectively [106,107,108,109], or miR-24 that regulates *PAK4*, *GATA2* [110], and *eNOS* [111], suggesting the use of encapsuled antagomiR as therapeutic strategy to improve angiogenesis and heart function after MI. 

In several studies, exosomes have been used as carriers delivering miRNAs, as mentioned above. Compelling evidence have shown that secreted exosomes from myocardial cells containing miR-125b, miR-126, miR-25-3p, miR-144, or miR-146a are associated with ischemic cardiac repair [23,24,112,113,114]. For example, exosomes derived from the peripheral serum of patients with acute MI containing significant amount of miR-126-3p, as compared to exosome isolated from healthy volunteers [115], efficiently increased ECs proliferation and microvessel sprouting by the activation of mTORC1/HIF-1α axis [115]. Similarly, exosomes derived from DEXs dendritic cells, which infiltrated into infarcted area following MI, overexpress miR-494-3p that enhanced cardiac microvascular ECs tube formation in vitro and angiogenesis, as demonstrated by significant increase of CD31+ marked cells in the infarcted myocardium [116]. Furthermore, the injection of exosomes derived from cardiac progenitor cells containing miR-322 in MI mice reduced the infarct size, as well as increased capillary density in heart tissue. These exosomes increased HUVEC tube formation and migration by increasing Nox2-derived ROS in vitro [117]. In a similar way, MSC-derived extracellular vesicles (MSC-EVs) enriched with miR-210 inhibited cardiac fibrosis, activated ECs and increased the number of capillaries in the peri-infarct regions of the post-MI mice heart [118]. The effect of MSC-EVs overexpressing miR-210 was induced by the inhibition of Efna3, a GPI-anchored membrane protein, as demonstrated in HUVEC [118]. This cardioprotective role of miR-210 was supported by another study which confirmed that overexpressing miR-210 promoted angiogenesis in adult transgenic mice following MI [119]. Another report investigated cardiac telocyte interstitial cells and ECs communication demonstrating that the delivery of cardiac telocytes to infarcted hearts increased vessels density thanks to the release of exosomal miR-21-5p, which modulated cell death inducing p53 target 1 (*Cdip1*) gene and downregulates caspase-3 expression [120].
ijms-24-12298-t001_Table 1Table 1miRNAs, targets genes, and pathways involved in post-myocardial infarction angiogenesis activated in endothelial cells. ↑: promotes; ↓: prevents. *CatK*: cathepsin K; *CUL2*: cullin 2; MACs: Myeloid angiogenic cells; *Efna3*: ephrin A3; *eNOS*: endothelial NO synthase; *HIF-1α*: hypoxia-inducible factor 1α; *IGF-IR*: insulin-like growth factor receptor 1; *PDK-1*: phosphoinositide-dependent protein kinase 1; *PIK3R2*: PI3K regulatory subunit 2; *PTEN*: phosphatase and tensin homolog; *PAK4*: p21 activated kinases 4; *ITGA5*: integrin subunit α 5; miRNA: microRNA; MSCs: mesenchymal stem cells; MSC-EVs: mesenchymal stem cells extra vesicles; *SPRED1*: sprouty-related protein 1; *SuFu*: suppressor of fused; *VEGF*: vascular endothelial growth factor.miRNAReleased or Delivered byTarget GenesPromotes/PreventsReferencesmiR-322/miR-424ECs*CUL2/**VEGF/GLUT1/EPO/HIF-1α*↑[94]miR-19a-3pCardiomyocytes*HIF-1α*↓[95]miR-126-3pMSCs/exosomes*PIK3R2/SPRED1/VEGF/bFGF/DLL4/**mTORC1/HIF-1α*↑[97,98]EPCs*VEGF-A/IL-10/IL-3/IGF-1/angiogenin*↑[99,115]miR126/miR-130aMACs*SPRED1*↑[100]miR-130aLentivirus*VEGF/HoxA5/AKT*↑[101]*PTEN*↓miR-375LNA anti-miR-375*PDK-1/AKT*↓[102]miR-21-5pCardiac stromal cells*PTEN/AKT*↓/↑[103]miR-499-5pTanshinone IIA administration*PTEN/VEGF/angiotensin-1*↑[104]miR-210MSC-EVs; transgenic mice*Efna3/VEGF*↑[118,119]miR-378CD34^+^ progenitor cells*SuFu/Fus-1*↑[105]miR-185-5pECs*CatK*↓[107]miR-143Cardiomyocytes*IGF-1R*↓[108]miR-92aECs*ITGA5*↓[109]miR-24encapsuled antagomiR*PAK4, GATA, eNOS*↓[110,111]


Altogether, these studies suggest that pro- or anti-angiogenic miRNAs target multiple signaling pathways involved in angiogenesis. Nevertheless, the reason why the ischemic insult can stimulate miRNAs with the opposite effect is still undiscovered, which is worth to be clarified.

## 7. Conclusions

Neovascularization within the infarcted tissue is an integral critical component of the cardiac remodeling process. The formation of the new dense capillary network would favor gas exchange, nutrient diffusion, and waste removal, which would attenuate cardiac myocyte dysfunction in the peri-infarcted zone. Understanding the complex mechanisms behind angiogenesis after MI may create multiple therapeutic opportunities. As summarized in Figure 1, in this review, we briefly highlight multiple signaling events implicated in this process, including classical pathways Notch, Wnt, and PI3K, and we discussed the role of endothelial progenitor cells [Ca^2+^]_i_ and miRNAs that regulate a wide range of genes implicated in EC proliferation, migration, and tube formation, the hallmarks of angiogenesis. Several miRNAs might be attractive candidates for modulating certain therapeutic targets in MI, although more work is still required to explore new targets and to develop efficient delivery vehicles into the infarcted heart.

## Figures and Tables

**Figure 1 ijms-24-12298-f001:**
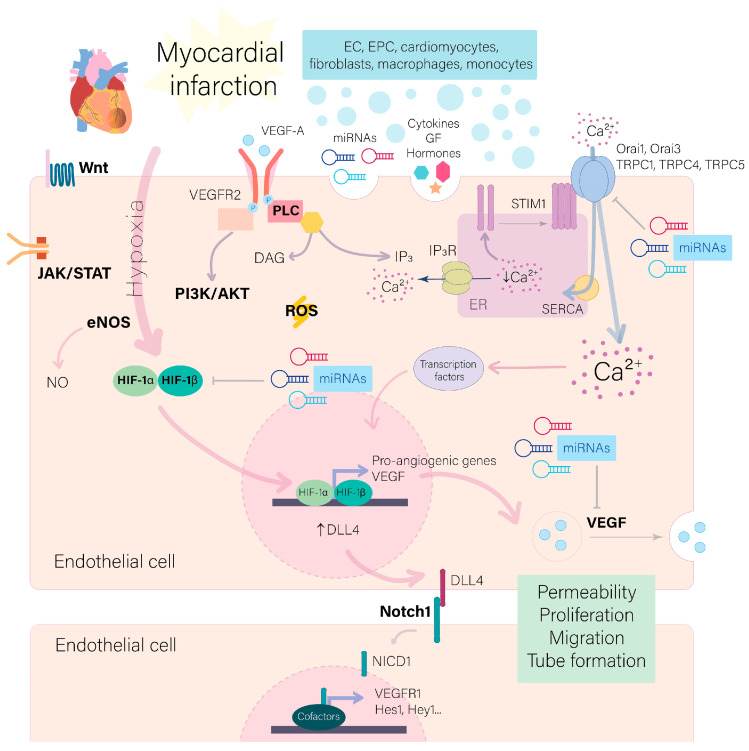
Signaling pathways activated in Endothelial Cells (ECs) after Myocardial Infarction (MI). During MI, ischemic cardiac tissue evokes the activation of HIF-1α, ROS, eNOS, Wnt, and JAK/STAT pathways in EC, which trigger the transcription of pro-angiogenic genes. Different heart cell populations (EC, EPC, cardiomyocytes, fibroblasts, macrophages) produce a high number of autocrine signals, including cytokines, growth factors such as VEGF, and non-coding RNAs, miRNAs, which regulate EC functions to rescue myocardium healing. VEGF induces the increase in intracellular Ca^2+^ concentration via store-operated Ca^2+^ entry (SOCE) through Orai and TRPC channels, which activates different transcription factors essential to ECs activity. VEGF increases DLL4 expression in EC, polarizing it to tip cell phenotype. In neighbor ECs, DLL4 activates Notch1 signaling, promoting the EC switch to stalk phenotype. Altogether, these events promote an increase in EC permeability, as well as in their proliferation, migration, and tube formation ability, hallmarks of angiogenesis. EPC: endothelial progenitor cells; ER: endoplasmic reticulum; DAG: directed acyclic graphs; DLL4: delta-like 4; eNOS: endothelial nitric oxide synthase; GF: growth factors; IP_3_: 1,4,5-trisphosphate; IP_3_R: IP_3_ receptor; miRNAs: microRNAs; NICD: Notch intracellular domain; PLC: phospholipase C; ROS: reactive oxygen species; SERCA: sarcoendoplasmic reticulum (SR) calcium transport ATPase; VEGF: vascular endothelial growth factor; VEGFR1: VEGF receptor 1; VEGFR2: VEGF receptor 2.

**Figure 2 ijms-24-12298-f002:**
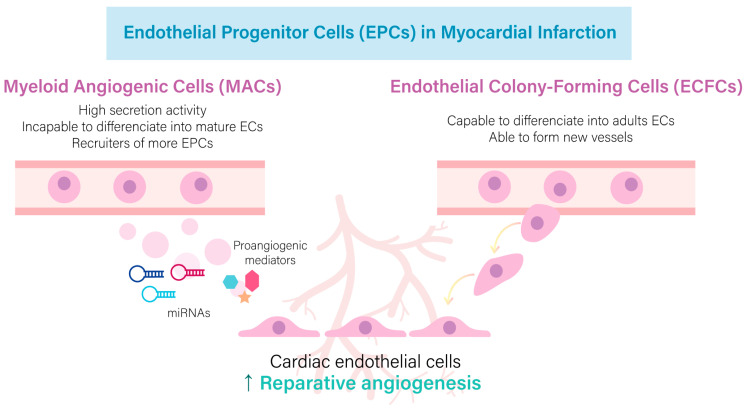
Endothelial Progenitor Cells (EPCs) are essential in post-ischemic angiogenesis. Both types of EPCs, Myeloid angiogenic cells (MACs) and endothelial colony forming (ECFCs), are actively involved in reparative neovascularization after myocardial infarction (MI). MACs are key regulators for EPCs recruitment as they secrete a wide range of pro-angiogenic factors, as well as microRNAs (miRNAs), to stimulate cardiac endothelial cells (ECs). Circulating and resident ECFCs can differentiate into mature ECs and take part in new blood vessel formation.

## Data Availability

Not applicable.

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
