# Peer review of "New Insights into the Reparative Angiogenesis after Myocardial Infarction"

_ijms, 2023, doi:10.3390/ijms241512298_

Round 1

Reviewer 1 Report (Previous Reviewer 2)

The authors have accepted my suggestions and I believe that it is a valuable review on the matter,

Author Response

Thank you for your constructive comments.

Reviewer 2 Report (Previous Reviewer 1)

First of all, I could not find the point-by-point answers of the authors to my previous comments. 

Second, I still could not find solid novel insights into reparative angiogenesis after MI after reading this paper, of note, except for some parts of the work. 

Third, I strongly disagree with the authors' statement in paragraph #2 that ECFCs contribute to de novo vessel formation after the onset of MI. We have to know clearly that ECFC appears after 14-21 days of cell culture and its existence in circulating blood is not proven. Moreover, contribution after MI has not been experimentally shown only shown culture cell-derived ECFC which is not the same as circulating cells, and its presence in circulation is still unknown. Every claim must be scientifically proved and cited properly. 

Fourth, according to the previous consensus, there is no term as called "early outgrowth cells", istead myeloid angiogenic cells should be used in manuscripts. 

Fifth, my previous question was not addressed. I recommended authors define miR target cells in the heart. 

First of all, I could not find the point-by-point answers of the authors to my previous comments. 

Second, I still could not find solid novel insights into reparative angiogenesis after MI after reading this paper, of note, except for some parts of the work. 

Third, I strongly disagree with the authors' statement in paragraph #2 that ECFCs contribute to de novo vessel formation after the onset of MI. We have to know clearly that ECFC appears after 14-21 days of cell culture and its existence in circulating blood is not proven. Moreover, contribution after MI has not been experimentally shown only shown culture cell-derived ECFC which is not the same as circulating cells, and its presence in circulation is still unknown. Every claim must be scientifically proved and cited properly. 

Fourth, according to the previous consensus, there is no term as called "early outgrowth cells", istead myeloid angiogenic cells should be used in manuscripts. Consequently, Fig two is not correct.

Fifth, my previous question was not addressed. I recommended authors define miR target cells in the heart. 

Author Response

First of all, I could not find the point-by-point answers of the authors to my previous comments. 

In the original resubmission we did uploaded our answers to your previous queries. This issue should be resolved and answered by the editorial team. Anyway, I’ve attached our original answers to your previous questions below your new comments.

Second, I still could not find solid novel insights into reparative angiogenesis after MI after reading this paper, of note, except for some parts of the work. 

Sorry for not reaching your expectations. We have reviewed many recent papers on reparative angiogenesis and discussed new findings in the field as highlighted in this resubmission. We sincerely believe that many parts of this review may be of interest to the scientific community.

Third, I strongly disagree with the authors' statement in paragraph #2 that ECFCs contribute to de novo vessel formation after the onset of MI. We have to know clearly that ECFC appears after 14-21 days of cell culture and its existence in circulating blood is not proven. Moreover, contribution after MI has not been experimentally shown only shown culture cell-derived ECFC which is not the same as circulating cells, and its presence in circulation is still unknown. Every claim must be scientifically proved and cited properly. 

We respect your opinion regarding the potential role of ECFCs in MI angiogenesis. We have gone through the manuscript again and we didn't find what you claim. We didn't discuss whether or not circulating ECFCs after MI by themselves can promote angiogenesis after MI. In particular, we discussed some papers that showed that ECFC administration under different conditions, e.g. stimulated by calcium silicate bioceramics (line 145, page 4) or in combination with MSCs (line 129 , page 4), can be used to resolve MI. We strongly believe that our claims are scientifically supported by consistent and well-designed experiments that we have correctly cited in these two recently well published papers: Yu, B. et al. Nat Commun 2023, 14, 2094; and Tripathi, H. et al. Stem Cell Rev Rep 2023, 19.

Fourth, according to the previous consensus, there is no term as called "early outgrowth cells", istead myeloid angiogenic cells should be used in manuscripts. 

Upon your suggestion myeloid angiogenic cells is used instead of EOC in the maon text and figure.

Fifth, my previous question was not addressed. I recommended authors define miR target cells in the heart. 

As shown in table legend, only Endothelial cells were the target for displayed miRNAs.

Here is what we answered in our original version:

“Following your advice and to facilitate reading we specified which kind of endothelial cells were used in the studies discussed in this review, even if in this report we mainly focused on miRs targeting genes in endothelial cells, as added now in the title of table 1. For this reason, we removed the information related to other cell types, as miRNA regulation of SOCs in immune cells. In addition, we detailed the mechanism of miRNAs, their genes target and related signaling pathway, as highlighted in distinct parts of miRNA’s section (Pages 8-10). In this section, we summarized the information about miRNAs which regulate the expression of HIF-1α (Pages 7-8, Lines 250-273); secondly, we talked about miRNAs which modulate VEGF or VEGF-related pathways (Pages 8-9, Lines 274-316); third, we discussed the role of miRNAs related to non-classical angiogenic pathways (Page 9, Lines 317-324); and finally, we gave an overview of articles focused on miRNAs carried out by exosomes which regulate different pathways involved in the to post-MI neovascularization (Pages 9-10; Lines 325-350).”

-------------------------------------------------------------------------------------------------------------------------

Sent July 3rd 2023

Answers to reviewer nº 1 comments:

We wish to thank the reviewer for his/her critiques and comments that certainly helped us to improve this revised version. We made significant changes and updated references as requested. We Hope that our valuable reviewer will be satisfied with this revised version. Please find below point-by-point our answers to your comments

1.- Authors using outdated references; for instance, in lines 61-62, authors state that EPC contributes following MI and cited references almost two decades ago. Recent investigation demonstrates no contribution following MI. Fujisawa et al. Circulation. Which type of EPC did the authors discuss?

First, we would like to apologize for using some old references. Actually, the contribution of EPCs to myocardial infarction remains a matter of debate, although many recent reports suggest that they are important agents in cardiac repair and angiogenesis, as their paracrine action improves the microenvironment after myocardial infarction, thereby driving cardiac remodeling and improving cardiac function as reviewed in https://doi.org/10.3389/fcvm.2021.717536.

We have read and included comments regarding the interesting work of Fujisawa et al. (https://doi.org/10.1161/CIRCULATIONAHA.119.042351) who investigated the origin of EPCs involved in vascular regeneration. This study concluded that endogenous neovascularization in the heart is driven by tissue resident EPCs without a direct contribution from bone marrow cells. Therefore, this study does not rule out the role of EPCs in post-infarction myocardial angiogenesis. In addition, Fujisawa et al agree with the work carried out by Li et al. (doi: 10.1093/eurheartj/ehz305) which revealed that, using single-cell transcriptome analyses, resident endothelial cells are the main responsible of angiogenesis after MI.

In this revised version of our manuscript, we provided recent references (11-17) and added details on the role of EPCs in angiogenesis in myocardial infarction, as highlighted in Page 2, line 64.

2.- Angiogenesis following myocardial infarction is a complex process, and various molecules ensemble work to initiate vessel growth. Authors tried to describe Ca2+ signaling importance following MI, but in reality, only six sentences in paragraph 2 or ref# 34,35,36 are related to the MI. Other references are not related to the MI. Since the review theme and discussing topics are novel angiogenesis mechanisms, we must understand that it should not deviate too much from the initial goal. Page 4 must be rewritten according to the current concept.

Thank you for your wise advice, we totally agree with you. In fact, our original idea was to describe the proangiogenic role of Ca2+ signaling in MI. However, the information regarding Ca2+ channel activation of endothelial cells in the context of cardiac infarction and angiogenesis is quite limited. Most reports only used ischemia and reperfusion in vitro protocol or other ischemic models, especially in the hind limbs, to investigate the activity of these channels in neovascularization. We agree that this might divert readers' attention from the main objective of this review. Therefore, following your suggestion, we rewrote this part of the manuscript and provided more details on the involvement of TRPC1 and 5 channels in angiogenesis under MI. We added this information on Pages 4-5 (Lines 140-169).

3.- Paragraph 3. Notch signaling. Notch signaling following MI is well studied, and some parts lost the novelty due to many review papers already discussed regarding Notch signaling (Int. J. Mol. Sci. 2022, 23(20), https://doi.org/10.1080/14728222.2019.1641198.

Since NOTCH and VEGF are the most studied angiogenesis pathways, we intend to give a brief overview of their signaling steps, not a deep discussion about the pathways, but we agree that both pathways have already been discussed in detail in other recent reviews. In this sense, we have re-edited this part by removing those widely known knowledge, and we added a recent article (doi: 10.1002/jcb.27032; 10.3389/fcell.2020.00011) about the role of Notch1 in neovascularization after MI.

4.- The most critical comment is the manuscript emphasized the VEGF pathway-dependent angiogenesis following Mi. There are a lot of genes that are responsible for angiogenesis besides VEGF. We highly encourage describing the classical and non-classical angiogenesis process after MI.

As mentioned in point 3, we revised this part and provided new section dealing with the role of reactive oxygen species (page 3, lines 104-119), as part of non-canonical pathways in angiogenesis, and more information about the role of other signaling pathways such as Wnt, PI3K, JAK/STAT (page 6, lines 207-223) in this reparative mechanism. In addition, we detailed the regulation of non-classical pro-angiogenic pathways in the section on miRNAs (Page 9, Lines 317-324).

5.- Please try to summarise the miRs depending on target cells. Just listing miRs is not proper; we want to have the mechanism of activation miRs to target cells, e.g., EC, cardiomyocyte, resident macrophages, etc.

Following your advice and to facilitate reading we specified which kind of endothelial cells were used in the studies discussed in this review, even if in this report we mainly focused on miRs targeting genes in endothelial cells, as added now in the title of table 1. For this reason, we removed the information related to other cell types, as miRNA regulation of SOCs in immune cells. In addition, we detailed the mechanism of miRNAs, their genes target and related signaling pathway, as highlighted in distinct parts of miRNA’s section (Pages 8-10). In this section, we summarized the information about miRNAs which regulate the expression of HIF-1α (Pages 7-8, Lines 250-273); secondly, we talked about miRNAs which modulate VEGF or VEGF-related pathways (Pages 8-9, Lines 274-316); third, we discussed the role of miRNAs related to non-classical angiogenic pathways (Page 9, Lines 317-324); and finally, we gave an overview of articles focused on miRNAs carried out by exosomes which regulate different pathways involved in the to post-MI neovascularization (Pages 9-10; Lines 325-350).

This manuscript is a resubmission of an earlier submission. The following is a list of the peer review reports and author responses from that submission.

Round 1

Reviewer 1 Report

Galeano-Otero et al. attempted to review the novel reparative angiogenesis following myocardial infarction. However, several shortcomings must be improved.

1) Authors using outdated references; for instance, in lines 61-62, authors state that EPC contributes following MI and cited references almost two decades ago. Recent investigation demonstrates no contribution following MI. Fujisawa et al. Circulation. Which type of EPC did the authors discuss?

2) Angiogenesis following myocardial infarction is a complex process, and various molecules ensemble work to initiate vessel growth. Authors tried to describe Ca2+ signaling importance following MI, but in reality, only six sentences in paragraph 2 or ref# 34,35,36 are related to the MI. Other references are not related to the MI. Since the review theme and discussing topics are novel angiogenesis mechanisms, we must understand that it should not deviate too much from the initial goal. Page 4 must be rewritten according to the current concept. 

3)Paragraph 3. Notch signaling. Notch signaling following MI is well studied, and some parts lost the novelty due to many review papers already discussed regarding Notch signaling (Int. J. Mol. Sci. 202223(20), https://doi.org/10.1080/14728222.2019.1641198. 

4) The most critical comment is the manuscript emphasized the VEGF pathway-dependent angiogenesis following Mi. There are a lot of genes that are responsible for angiogenesis besides VEGF. We highly encourage describing the classical and non-classical angiogenesis process after MI. 

5) Please try to summarise the miRs depending on target cells. Just listing miRs is not proper; we want to have the mechanism of activation miRs to target cells, e.g., EC, cardiomyocyte, resident macrophages, etc. 

Author Response

Galeano-Otero et al. attempted to review the novel reparative angiogenesis following myocardial infarction. However, several shortcomings must be improved.

Answers to reviewer nº 1 comments:

We wish to thank the reviewer for his/her critiques and comments that certainly helped us to improve this revised version. We made significant changes and updated references as requested. We Hope that our valuable reviewer will be satisfied with this revised version. Please find below point-by-point our answers to your comments

1.- Authors using outdated references; for instance, in lines 61-62, authors state that EPC contributes following MI and cited references almost two decades ago. Recent investigation demonstrates no contribution following MI. Fujisawa et al. Circulation. Which type of EPC did the authors discuss?

First, we would like to apologize for using some old references. Actually, the contribution of EPCs to myocardial infarction remains a matter of debate, although many recent reports suggest that they are important agents in cardiac repair and angiogenesis, as their paracrine action improves the microenvironment after myocardial infarction, thereby driving cardiac remodeling and improving cardiac function as reviewed in https://doi.org/10.3389/fcvm.2021.717536.

We have read and included comments regarding the interesting work of Fujisawa et al. (https://doi.org/10.1161/CIRCULATIONAHA.119.042351) who investigated the origin of EPCs involved in vascular regeneration. This study concluded that endogenous neovascularization in the heart is driven by tissue resident EPCs without a direct contribution from bone marrow cells. Therefore, this study does not rule out the role of EPCs in post-infarction myocardial angiogenesis. In addition, Fujisawa et al agree with the work carried out by Li et al. (doi: 10.1093/eurheartj/ehz305) which revealed that, using single-cell transcriptome analyses, resident endothelial cells are the main responsible of angiogenesis after MI.

In this revised version of our manuscript, we provided recent references (11-17) and added details on the role of EPCs in angiogenesis in myocardial infarction, as highlighted in Page 2, line 64.

2.- Angiogenesis following myocardial infarction is a complex process, and various molecules ensemble work to initiate vessel growth. Authors tried to describe Ca2+ signaling importance following MI, but in reality, only six sentences in paragraph 2 or ref# 34,35,36 are related to the MI. Other references are not related to the MI. Since the review theme and discussing topics are novel angiogenesis mechanisms, we must understand that it should not deviate too much from the initial goal. Page 4 must be rewritten according to the current concept.

Thank you for your wise advice, we totally agree with you. In fact, our original idea was to describe the proangiogenic role of Ca2+ signaling in MI. However, the information regarding Ca2+ channel activation of endothelial cells in the context of cardiac infarction and angiogenesis is quite limited. Most reports only used ischemia and reperfusion in vitro protocol or other ischemic models, especially in the hind limbs, to investigate the activity of these channels in neovascularization. We agree that this might divert readers' attention from the main objective of this review. Therefore, following your suggestion, we rewrote this part of the manuscript and provided more details on the involvement of TRPC1 and 5 channels in angiogenesis under MI. We added this information on Pages 4-5 (Lines 140-169).

3.- Paragraph 3. Notch signaling. Notch signaling following MI is well studied, and some parts lost the novelty due to many review papers already discussed regarding Notch signaling (Int. J. Mol. Sci. 2022, 23(20), https://doi.org/10.1080/14728222.2019.1641198.

Since NOTCH and VEGF are the most studied angiogenesis pathways, we intend to give a brief overview of their signaling steps, not a deep discussion about the pathways, but we agree that both pathways have already been discussed in detail in other recent reviews. In this sense, we have re-edited this part by removing those widely known knowledge, and we added a recent article (doi: 10.1002/jcb.27032; 10.3389/fcell.2020.00011) about the role of Notch1 in neovascularization after MI.

4.- The most critical comment is the manuscript emphasized the VEGF pathway-dependent angiogenesis following Mi. There are a lot of genes that are responsible for angiogenesis besides VEGF. We highly encourage describing the classical and non-classical angiogenesis process after MI.

As mentioned in point 3, we revised this part and provided new section dealing with the role of reactive oxygen species (page 3, lines 104-119), as part of non-canonical pathways in angiogenesis, and more information about the role of other signaling pathways such as Wnt, PI3K, JAK/STAT (page 6, lines 207-223) in this reparative mechanism. In addition, we detailed the regulation of non-classical pro-angiogenic pathways in the section on miRNAs (Page 9, Lines 317-324).

5.- Please try to summarise the miRs depending on target cells. Just listing miRs is not proper; we want to have the mechanism of activation miRs to target cells, e.g., EC, cardiomyocyte, resident macrophages, etc.

Following your advice and to facilitate reading we specified which kind of endothelial cells were used in the studies discussed in this review, even if in this report we mainly focused on miRs targeting genes in endothelial cells, as added now in the title of table 1. For this reason, we removed the information related to other cell types, as miRNA regulation of SOCs in immune cells.

In addition, we detailed the mechanism of miRNAs, their genes target and related signaling pathway, as highlighted in distinct parts of miRNA’s section (Pages 8-10). In this section, we summarized the information about miRNAs which regulate the expression of HIF-1α (Pages 7-8, Lines 250-273); secondly, we talked about miRNAs which modulate VEGF or VEGF-related pathways (Pages 8-9, Lines 274-316); third, we discussed the role of miRNAs related to non-classical angiogenic pathways (Page 9, Lines 317-324); and finally, we gave an overview of articles focused on miRNAs carried out by exosomes which regulate different pathways involved in the to post-MI neovascularization (Pages 9-10; Lines 325-350).

Reviewer 2 Report

The work ‘New insights into the reparative angiogenesis after myocardial 2 infarction’ is a well written paper with nice figures. It brings a nice review on the subject and should be accepted after changes.

-          One cannot address MI without addressing ROS. That is key in the activation of several mechanisms herein and related with angiogenesis (https://pubmed.ncbi.nlm.nih.gov/19758416/)

-          Also related with the prior, the role of inflammation and mainly NADPH oxidase needs to be discussed in the light of what this review aims to bring

-          Antioxidants (Basel). 2019 Jan 10;8(1):18. doi: 10.3390/antiox8010018

-          https://pubmed.ncbi.nlm.nih.gov/22038056/

-          NO is mostly ignored and cannot be overlooked. Some data needs to be placed in perspective regarding M/I and even angiogenesis

Minor:

More figures of each section and a better quality figure is mandatory

Author Response

The work ‘New insights into the reparative angiogenesis after myocardial 2 infarction’ is a well written paper with nice figures. It brings a nice review on the subject and should be accepted after changes.

We wish to thank this reviewer for his/her critiques and comments which certainly helped us to improve this revised version. I Hope that our valuable reviewer will be satisfied with this revised version. We made significant changes and answered to your request as detailed:

1.- One cannot address MI without addressing ROS. That is key in the activation of several mechanisms herein and related with angiogenesis (https://pubmed.ncbi.nlm.nih.gov/19758416/).

2.- Also related with the prior, the role of inflammation and mainly NADPH oxidase needs to be discussed in the light of what this review aims to bring Antioxidants (Basel). 2019 Jan 10;8(1):18. doi: 10.3390/antiox8010018. https://pubmed.ncbi.nlm.nih.gov/22038056/.

3.- NO is mostly ignored and cannot be overlooked. Some data needs to be placed in perspective regarding M/I and even angiogenesis

You are correct. ROS, NADPH and NO are fundamental to understanding the development of myocardial infarction disease. As this valuable reviewer is well aware, angiogenesis after myocardial infarction is a complex process, involving the activation of several independent signaling pathways. Herein, upon your request we have added data on these relevants points as detailed in the main text.

We have added a new paragraph on page 3 in which we discuss current knowledge about ROS activation in endothelial cells and angiogenesis in myocardial infarction (page 3, line 104) and discuss the role of NADPH oxidase in angiogenesis in myocardial infarction (page 3, line 108; and page 10, line 336). This is also the case for the role of nitric oxide, which is now addressed on page 3, line 113.

Minor:

More figures of each section and a better quality figure is mandatory

We uploaded a new figure with high quality and added new figure for graphical abstract.

Round 2

Reviewer 1 Report

My comments from 1st revision were not addressed. In this form, it is not suitable for publication. 

Reviewer 2 Report

New figures are required on the complex mechanisms described. They are mandatory in a well successfull review.